# Perinatal outcome of vaginal breech delivery in Orotta National Referral Teaching Hospital, Eritrea, 2024; a case control study

Okbu Frezgi [1]*, Berhe Tesfai[1], Abraham Amanuel[1], Khalid Hussein[1], Hagos Teckle[2], Hailemichael Gebremariam[3], Andebrhan Tewelde[4]

1 Department of Obstetrics and Gynecology, Orotta College of Medicine and Health Science, Orotta National Referral Teaching Hospital, Ministry of Health, Asmara, Eritrea, 2 Department of Pediatrics, Orotta College of Medicine and Health Science, Orotta National Referral Pediatrics Hospital, Ministry of Health, Asmara, Eritrea, 3 Dekemhare Hospital, Zoba Debub, Ministry of Health, Dekemhare, Eritrea, 4 Ministry of Health, Monitoring and Evaluation Division, Asmara, Eritrea

* lurgewra@gmail.com

## Abstract

### Introduction

Delivery of fetus in breech presentation is a controversial topic in obstetrics and it had been the subject of debate since 1980s. As experts with abilities and know-how to perform vaginal breech delivery have decreased, and medico legal concerns increased, some physicians endorse cesarean section for breech presentation at term. The aim of this study was to identify the perinatal outcome in vaginal breech deliveries.

### Methodology

This study was a retrospective case control study of mothers delivered vaginally between January 1, 2019 and December 31, 2023 in Orotta National Referral Teaching Hospital. Mothers who gave vaginal breech delivery (VBD) were enrolled as cases and two successive vertex vaginal deliveries (VVD) as controls.

### Results

A total of 1919 patient records were analyzed, of which 641(33.6%) were cases and 1278 (66.4%) controls. The mean age and parity of the mothers was 28.98 (SD = 5.56) and 2.18 (SD = 1.96) respectively. History of previous stillbirth was documented in 0.4% and multiple pregnancies found in 10.2% of the study participants. The prevalence of stillbirth was 5% (1% in controls and 4% in cases). In multivariate analysis address from other zobas (AOR: 3.63, 95% CI: 2.17–6.09, p < 0.011), high gravidity (AOR: 1.62, 95% CI: 1.18–2.22, p < 0.001), low birth weight (AOR: 2.03, 95% CI: 1.42–2.90, p < 0.001), and multiple fetus (AOR: 4.83, 95% CI: 3.28–7.11,

**Data availability statement:** All necessary documents are included with the manuscript.

**Funding:** The author(s) received no specific funding for this work.

**Competing interests:** The authors have declared that no competing interests exist.

**Abbreviations:** AOR, Adjusted Odds Ratio; ANC, Antenatal Care; CS, Caesarean section; CI, Confidence Interval; COR, Crude Odds Ratio; IUFD, Intra Uterine Fetal Death; IUGR, Intra Uterine Growth Restriction; SD, Standard deviation; WHO, World Health Organization; VBD, Vaginal breech delivery; VVD, Vaginal vertex delivery.

p < 0.001, were associated with risk of having vaginal breech delivery. Primiparity (AOR: 0.25, 95% CI: 0.18–0.36 p < 0.001) and birth weight >3.4 kg (AOR: 0.61, 95% CI: 0.47–0.79, p < 0.001) were protective against having vaginal breech delivery. Vaginal breech deliveries was found to be associated low first minute Apgar (AOR: 14.95, 95% CI: 9.36–23.26, p < 0.001).

## Conclusion

Vaginal breech delivery was associated with low first minute Apgar. Address from others zobas, high gravidity, low birth weight, and multiple fetuses increases the risk of having Breech presentation, Vaginal breech delivery, Perinatal morbidity, Nulliparous vaginal breech delivery while primiparity and increasing birth weight were protective.

---

## Introduction

Advancing pregnancy declines the incidence of singleton fetus's breech presentation to around 3–4% [1]. The commonest cause of breech presentation is preterm delivery in which every fourth of all fetuses born extremely preterm are in breech presentation [2,3]. Previous breech presentation, uterine configuration abnormalities, abnormalities in placental location, multiparty, polyhydramnios, contracted pelvis, fetal anomalies, multiple gestation, short umbilical cord, and fetal growth restriction are also common causes of breech presentation [4,5]. Comparing to cephalic presentation, the outcomes for breech deliveries are worse, irrespective of the mode delivery [6].

Literature suggests that planned vaginal breech delivery (VBD) has higher perinatal complications, including intraventricular hemorrhage, seizures, low Apgar scores, brachial plexus injury, and neonatal death than planned caesarean section (CS) [7]. Neonatal mortality in breech presentation has continued to remain 3–5 times higher than that of cephalic presentation [8]. In two studies in Sub-Saharan countries, there was a strong association between vaginal breech delivery of singleton term pregnancies and feto-maternal morbidity, with newborns more likely to suffer from birth asphyxia [9]. Mothers with breech infants at term seek counseling regarding the safest delivery mode [10,11]. Counseling for term breech pregnancies often steers women towards CS and only addresses short-term risks to the baby [12].

Currently, the management of term breech presentation is the most controversial topics in obstetric and it has been the subject of debate since the 1980s [13]. In 2000, changes in clinical practice were introduced after a randomised multicentre collaborative study about how to deal with term breech delivery published by the authors of the Term Breech Trial Collaborative Group (TBT) and they concluded that elective CS offered better results than vaginal deliveries in full-term fetuses with breech presentation [14]. The authors of green top guideline suggest CS delivery for estimated fetal weight above 3.8 kg [15]. In Beijing China, breech presentations undergo CS, with the rate being as high as 90.68%, however, in Tibet, most breech presentations are still delivered vaginally [16].

According to Cunningham et al., if hydrocephaly is excluded, the head is flexed, the parietal diameter is less than 10 cm, a footling breech is ruled out, and the fetus is estimated to be of average weight, a VBD can be anticipated but certain proof that caesarean breech delivery improves neonatal outcome is lacking [17]. As number of practitioners with the skills and experience to perform vaginal breech delivery has decreased, there is a trend to perform caesarean delivery for term singleton fetuses in a breech presentation in developed world [18]. Obstetrician–gynecologists and other obstetric care providers should offer external cephalic version as an alternative to planned CS delivery for a woman who has a term singleton breech fetus [19]. In resources limited rural areas, a proper management plan before the onset of labour is often not achievable and local evidence based guidelines to recommend the most suitable delivery mode for every individual patient is warranted [20].

Determining the outcome of vaginal breech delivery is very crucial for deciding the mode of delivery for better perinatal and maternal outcomes. The general objective of this study was to identify the determinants of perinatal outcome of vaginal breech deliveries in Orotta National Referral Teaching Hospital.

## Methodology

### Study design

This was a retrospective case control study with medical records review of mothers delivered vaginally from January 1, 2019 to December 31, 2023 in Orotta National Referral Teaching Hospital. The data were accessed from 1st April –30th June and authors had accesses to individual participant's information.

### Study population and sampling method

All mothers who give birth vaginally during the study time were the study population. All VBD were considered as cases and two consecutive VVD were taken as controls.

### Inclusion and exclusion criteria

Delivery registers with complete information were included and neonates with a birth weight of < 1 kg, gross congenital anomalies, and instrumental vaginal deliveries were excluded from the study.

### Data collection and analysis

A specific data extraction tool was designed to retrieve data from the delivery register. Pilot study was conducted before starting the actual data extraction to ratify and modify the data collection tool to the context and objectives of the study. Data collection tool encompasses the socio-demographic characteristics of the patient (age, parity, gravidity, mode of delivery, year of delivery, and previous history of abortion or still birth) and the perinatal birth outcomes (sex, birth weight, 1st and fifth minute Apgar score, alive/stillbirth, referred to perinatal ICU).

Data was collected by physicians and the collected data were further checked for completeness and entered in MS excel with data cleaning and missed data was refilled by checking their register. Finally, the data was exported to SPSS version 26 for further analysis. The frequency and percent were determined and mean was calculated for continuous variables. Univariate and multivariate analysis was done to determine the association between the outcome and exposures of the variables and p-value < 0.05 was considered significant.

### Operational definitions

Abortion: Defined as a clinically recognized pregnancy loss before the 28th week of gestation.

Extreme prematurity: Delivery of fetus from 28–32 completed weeks.

### Ethical considerations

Ethical approval was obtained from the Ministry of Health Research Approval and Ethical Committee (reference number 30/03/2023) and respected authorities were informed. Informed consent was not sought as this was a secondary data. The personal identifiers of patients were coded and analyzed as aggregates. Patients didn't have any harm by using their data in this study.

## Results

### Socio-demographic characteristics and perinatal outcome of mothers delivered vaginally in Orotta National Referral Teaching Hospital

A total of 1919 patient records analyzed, of which 641(33.6%) were cases and 1278(66.4%) controls (Table 1). The mean age of the mothers was 28.98 (SD = 5.56) years of which 58% (25–35 years), 26% (<25 years), and 19% (>35 years). The mean parity was 2.18 (SD = 1.96) being distributed as 62.2% multiparous, 23.1% primiparous and 12.7% grandmultiparous. History of previous stillbirth was documented in 0.4%, one abortion in 14.6% and >1 abortion in 5.1%. Multiple pregnancy was found in 10.2% of the total deliveries and 0.2% had triplets. Majority of deliveries during a day take place from mid night to 8:00 am (35.5%) and the lowest from 8:00 am to 4:00 pm (30.5%). According to the months of a year the lowest delivery rate was documented on August (5.9%) and the highest was on January and October (9.5%) (Fig 1). The rate of VBD progressively decreases from 27.5% in 2019 to 13.7% in 2023.

### Perinatal outcome of mothers delivered vaginally from January 1, 2019 to December 31, 2023 at Orotta National Referral Teaching Hospital

The sex of the newborns was almost of 1:1 ratio. The mean birth weight of neonates was 3.08 kg, distributed as 57% (2.5–3.4 kg), 19% (≥ 3.5), 12% (1.5–2.4) and 2% (<1.5). The overall 1st minute Apgar was documented <7 in 20% of the neonates (cases 15.3% vs controls 4.5%) and 11% had < 7 Apgar at 5th minute (cases 8.8% vs controls 2.5%) which showed strong difference between cases and controls. Overall NICU admission was 9% (cases 6.14% vs controls 2.76%) with strong difference between cases and controls. The overall prevalence of stillbirth was 5% (1% in controls and 4% in cases) and vaginal breech delivery had a relationship with stillbirth (p-value <0.001).

### Univariate analysis of socio-demographic characteristics and perinatal outcome of mothers delivered vaginally

Increasing maternal age (COR: 2.08, 95% CI: 1.64–2.63, P < 0.001), address from other zobas (COR: 4.56, 95% CI: 1.87–2.83, P < 0.001) increasing gravidity (COR: 2.04, 95% CI: 1.55–2.70, P < 0.001), low birth weight (COR: 3.44, 95% CI: 2.56–4.62, P < 0.001), and multiple fetuses (COR: 5.61, 95% CI: 4.07–7.75, P < 0.001) were associated with increased risk to have vaginal breech delivery. Besides increasing birth weight protects risk to have vaginal breech delivery (COR: 0.73, 95% CI: 0.58–0.93, P < 0.001). Vaginal breech delivery associated with Apgar of <7 first and fives minute (COR: 11.60, 95% CI: 8.89–15.15, p < 0.001) and (COR: 9.18, 95% CI: 6.55–12.86, P < p < 0.001) respectively. Furthermore, the univariate analysis had showed vaginal breech delivery was associated with increased risk of having stillbirth (COR 9.18, 95% CI 5.51–15.31, P < 0.001) and NICU admission (COR 5.21, 95% CI 3.71–7.33, P < 0.001) (Table 2).

### Multivariate analysis of sociodemographic characteristics and perinatal outcome of mothers delivered vaginally

In multivariate analysis address from other zoba (AOR: 3.63, 95% CI: 2.17–6.09, p < 0.001), high gravidity (AOR: 1.62, 95% CI: 1.18–2.22, p < 0.001), low birth weight (AOR: 2.03, 95% CI: 1.42–2.90, p < 0.001), and multiple fetuses (AOR: 4.83, 95% CI: 3.28–7.11, p < 0.001) were associated with higher odds of having VBD. Primiparity (AOR: 0.25, 95% CI: 0.18–0.36, p < 0.001) and birth weight > 3.4 kg (AOR: 0.61, 95% CI: 0.47–0.79, p < 0.001) were protective against having

**Table 1. Sociodemographic characteristics and perinatal outcome of mothers delivered vaginally from January 1, 2019 to December 31, 2023 at Orotta National Referral Teaching Hospital. (n = 1,919).**

| Variables | Total deliveries N (%) | Mode of delivery | | P Value |
|---|---|---|---|---|
| | | VVD N (%) | VBD N (%) | |
| **Age** | | | | |
| 16-19 | 51 (3) | 39 | 12 | <0.001 |
| 20-34 | 1,513 (79) | 1,053 | 460 | |
| >34 | 355 (18) | 186 | 169 | |
| **Address** | | | | |
| Maekel | 1,814 (95) | 1,244 | 570 | <0.001 |
| Other zobas | 105 (5) | 34 | 71 | |
| **Gravida** | | | | |
| 1-4 | 1,384 (72) | 996 | 388 | <0.001 |
| >4 | 535 (28) | 282 | 253 | |
| **Parity** | | | | |
| 0 | 444 (23) | 360 | 84 | <0.001 |
| 1-4 | 1,232 (64) | 802 | 430 | |
| >4 | 243 (13) | 116 | 127 | |
| **Abortion** | | | | |
| 0 | 1,542 (80) | 1,058 | 484 | <0.001 |
| 1 | 280 (15) | 162 | 118 | |
| >1 | 97 (5) | 58 | 39 | |
| **History of still birth** | | | | |
| No | 1911 (99.6) | 1,275 | 636 | 0.127 |
| Yes | 8 (0.4) | 3 | 5 | |
| **Birth weight** | | | | |
| 1-1.4 kg | 46 (2) | 12 | 34 | <0.001 |
| 1.5-2.4 kg | 229 (12) | 91 | 138 | |
| 2.5-3.4 kg | 1,088 (57) | 755 | 333 | |
| >3.5 kg | 556 (29) | 420 | 136 | |
| **Twin** | | | | |
| No | 1,723 (90) | 1,219 | 504 | p<0.001 |
| Yes | 196 (10) | 59 | 137 | |
| **APGAR 1st minute** | | | | |
| <7 | 381 (20) | 87 | 294 | <0.001 |
| >=7 | 1538 (80) | 1,191 | 347 | |
| **APGAR 5th minute** | | | | |
| <7 | 217 (11) | 48 | 169 | <0.001 |
| >=7 | 1,702 (89) | 1,230 | 472 | |
| **Sex** | | | | |
| Female | 973 (51) | 637 | 336 | 0.319 |
| Male | 946 (49) | 641 | 305 | |
| **Alive or stillbirth** | | | | |
| Alive | 1,822 (95) | 1,259 | 563 | <0.001 |
| Stillbirth | 97 (5) | 19 | 78 | |
| **Refer to NICU** | | | | |
| No | 1,748 (91) | 1225 | 523 | <0.001 |
| Yes | 171 (9) | 53 | 118 | |

VVD, vaginal vertex delivery; VBD, vaginal breech delivery.

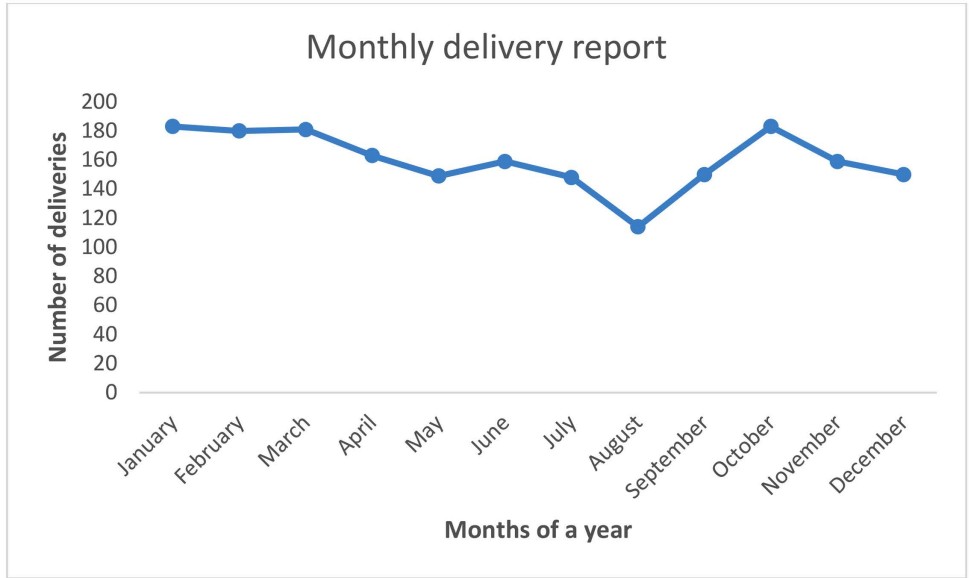

**Fig 1. Distribution of the rate vaginal deliveries from January 1, 2019 to December 31, 2023 at Orotta National Referral Teaching Hospital (n = 1,919).**

VBD. Besides, multivariate analysis showed, vaginal breech delivery was associated with increased risk of having low first minute Apgar (AOR: 14.95, 95% CI: 9.47–23.58, p < 0.001) (Table 3).

## Discussion

Appropriate route of delivery for fetuses on breech presentation has become a challenging issue in the last two decades. Based on results of several studies, personal experiences, and medico legal concerns, led some physicians to recommend CS delivery for breech presentation [4,10]. This study was designed to test this common clinical scenario and take a more detailed look at the safety and success of vaginal breech deliveries in resources limited setting. In this study the VBD progressively decline from 2019 during which time a residence program was reinstitute in the hospital and possibly this could be a reason for the declining rate of VBD. The study setting was in zoba maekel and the reason for the higher odds of having VBD in women's from others zoba was due to referral related intrapartum related complication.

Perinatal mortality in breech presentation has continued to remain 3–5 times higher than that of cephalic presentation [8]. Similarly, in our study VBD was associated with high rate of stillbirth compared to vertex vaginal deliveries (VVD 1%, vs VBD 4%). A study conducted by Kemfang Ngowa et. al in Cameroon also reported a significant perinatal mortality for breech deliveries [21]. Contrarily, a study conducted in France and Belgium stated that when strict criteria are met before and during labor, planned vaginal delivery of singleton fetuses in breech presentation at term remains a safe option that can be offered to women [22]. The difference between low-and middle income countries and high income countries in the perinatal outcome probably related to the difference in the service being given during antenatal and intrapartum care. In low resources setting the absence of well-defined selection criteria for VBD and relatively poorer experience of vaginal breech deliveries might probably increase the adverse pregnancy outcomes.

Increasing maternal age and multi-parity were found to be associated with increased VBD and better perinatal outcomes comparing to counterparts in bivariate analysis. A study conducted by U. Kielland-Kaisen et al stated that, perinatal morbidity and mortality was not significantly different in deliveries of nulliparous compared to multiparous [23]. In our study small number of nulliparous women tend to have VBD and this could be due to the direct CS delivery breech presentation

**Table 2. Univariate analysis of maternal demographic characteristics of mothers delivered vaginally from January 1, 2019 to December 31, 2023 at Orotta National Referral Teaching Hospital. (n = 1,919).**

| Variables | Mode of delivery | | COR | 95% CI | P-value |
|---|---|---|---|---|---|
| | VVD N (%) | VBD N (%) | | | |
| **Age** | | | | | |
| 16-19 | 39 (76) | 12 (24) | 0.703 | 0.37-1.36 | 0.295 |
| 20-34 (reference) | 1,053 (70) | 460 (30) | – | – | – |
| >34 | 186 (52) | 169 (48) | 2.08 | 1.64-2.63 | <0.001 |
| **Address (Zoba)** | | | | | |
| Maekel | 1,244 (64.82) | 570 (29.7) | – | | |
| Other Zobas | 34 (1.8) | 71 (3.7) | 4.56 | 2.99–6.94 | <0.001 |
| **Gravida** | | | | | |
| 1-4 | 996 (72) | 388 (28) | – | – | – |
| >4 | 282 (53) | 253 (47) | 2.03 | 1.87 - 2.83 | <0.001 |
| **Parity** | | | | | |
| 0 | 360 (81) | 84 (19) | 0.44 | 0.33 - 0.57 | <0.001 |
| 1-4 (reference) | 802 (65) | 430 (35) | – | – | – |
| >4 | 116 (48) | 127 (52) | 2.04 | 1.55 - 2.70 | <0.001 |
| **Abortion** | | | | | |
| 0 | 1,058 (83) | 484 (76) | – | – | – |
| 1 | 162 (13) | 118 (18) | 1.59 | 1.23 - 2.07 | <0.001 |
| >1 | 58 (5) | 39 (6) | 1.47 | 0.97 - 2.24 | 0.072 |
| **Birth weight** | | | | | |
| 1-1.4 kg | 12 (1) | 34 (5) | 6.42 | 3.29 - 12.56 | <0.001 |
| 1.5-2.4 kg | 91 (7) | 138 (22) | 3.44 | 2.56 - 4.62 | <0.001 |
| 2.5-3.4 kg (reference) | 755 (59) | 333 (52) | – | – | – |
| >3.4 kg | 420 (33) | 136 (21) | 0.73 | 0.58 - 0.93 | <0.001 |
| **Multiple fetus** | | | | | |
| No | 1,219 (63.5) | 504 (26.26) | – | – | |
| Yes | 59 (3.1) | 137 (7.2) | 5.61 | 4.069-7.753 | <0.001 |
| **APGAR 1st minute** | | | | | |
| >=7 | 1,191 (93) | 347 (54) | – | – | – |
| <7 | 87 (7) | 294 (46) | 11.60 | 8.89 - 15.15 | <0.001 |
| **APGAR 5th minute** | | | | | |
| >=7 | 1,230 (96) | 472 (74) | – | – | – |
| <7 | 48 (4) | 169 (26) | 9.18 | 6.55 - 12.86 | <0.001 |
| **Sex** | | | | | |
| Female | 637 (50) | 336 (52) | – | – | – |
| Male | 641 (50) | 305 (48) | 0.90 | 0.75 - 1.09 | 0.287 |
| **Alive or stillbirth** | | | | | |
| Alive | 1,259 (99) | 563 (88) | – | – | – |
| Stillbirth | 19 (1) | 78 (12) | 9.18 | 5.51 - 15.31 | <0.001 |
| **Refer to NICU** | | | | | |
| No | 1225 (96) | 523 (82) | – | – | – |
| Yes | 53 (4) | 118 (18) | 5.21 | 3.71 - 7.33 | <0.001 |

VVD, vaginal vertex delivery; VBD, vaginal breech delivery.

**Table 3. Multivariate analysis of vaginal breech delivery and perinatal outcome on mothers delivered vaginally from January 1, 2019 to December 31, 2023 at Orotta National Referral Teaching Hospital.**

| Variables | Mode of delivery | | AOR | 95% CI | P-value |
|---|---|---|---|---|---|
| | VVD N (%) | VBD N (%) | | | |
| **Age** | | | | | |
| 16-19 | 39 (76) | 12 (24) | 1.72 | 0.76 - 3.89 | 0.19 |
| 20-34 (reference) | 1,053 (70) | 460 (30) | – | – | – |
| >34 | 186 (52) | 169 (48) | 1.28 | 0.94 - 1.75 | 0.119 |
| **Address (Zoba)** | | | | | |
| Maekel | 1,244 (64.82) | 570 (29.7) | – | | |
| Other Zobas | 34 (1.8) | 71 (3.7) | 3.63 | 2.17–6.09 | <0.001 |
| **Gravida** | | | | | |
| 1-4 | 996 (72) | 388 (28) | | | |
| >4 | 282 (53) | 253 (47) | 1.62 | 1.18 - 2.22 | 0.003 |
| **Parity** | | | | | |
| 0 | 360 (81) | 84 (19) | 0.25 | 0.18 - 0.36 | <0.001 |
| 1-4 (reference) | 802 (65) | 430 (35) | – | – | – |
| >4 | 116 (48) | 127 (52) | 1.27 | 0.85 - 1.90 | 0.237 |
| **Birth weight** | | | | | |
| 1-1.4 kg | 12 (1) | 34 (5) | 1.61 | 0.72 - 3.62 | 0.244 |
| 1.5-2.4 kg | 91 (7) | 138 (22) | 2.03 | 1.42 - 2.90 | <0.001 |
| 2.5-3.4 kg (reference) | 755 (59) | 333 (52) | – | – | – |
| >3.4 kg | 420 (33) | 136 (21) | 0.61 | 0.47 - 0.79 | <0.001 |
| **Multiple Fetus** | | | | | |
| No | 1,219 (63.5) | 504 (26.26) | – | – | |
| Yes | 59 (3.1) | 137 (7.2) | 4.83 | 3.28–7.11 | <0.001 |
| **APGAR 1st minute** | | | | | |
| >=7 | 1,191 (93) | 347 (54) | – | – | – |
| <7 | 87 (7) | 294 (46) | 14.75 | 9.36 - 23.26 | p<0.001 |
| **APGAR 5th minute** | | | | | |
| >=7 | 1,230 (96) | 472 (74) | – | – | – |
| <7 | 48 (4) | 169 (26) | 0.81 | 0.43 - 1.52 | 0.516 |
| **Alive or stillbirth** | | | | | |
| Alive | 1,259 (99) | 563 (88) | – | – | – |
| Stillbirth | 19 (1) | 78 (12) | 1.26 | 0.56 - 2.84 | 0.575 |
| **Refer to NICU** | | | | | |
| No | 1225 (96) | 523 (82) | – | – | – |
| Yes | 53 (4) | 118 (18) | 0.86 | 0.50 - 1.47 | 0.588 |

VVD, vaginal vertex delivery; VBD, vaginal breech delivery.

incorporated in the local guideline. This guideline was adopted based on clinical experience of senior obstetricians in the association of poor perinatal outcome and VBD in nulliparous. Mothers with breech presentation should be identified during antenatal care, and mode of delivery should be planned after trial of external cephalic version if indicated and final decision should be based on attitude of the fetus, pelvic adequacy, and estimated fetal weight.

This study revealed that, low birth weight was found to be risk factor for VBD while birth weight of > 3.4 kg was protective which was consistent with other study conducted in another similar setting [15]. Higher NICU admission was also

documented in our report, likewise to a study conducted by Hashim M et al in Sudan which showed low 5-min Apgar scores, and admission to the neonatal care unit [24]. A study done by Qaiser Javed Iqbal et al also reported poor Apgar score and higher NICU admission of vaginally delivered neonates [25]. Our study found a significant low 1st min Apgar in VBD similar to study conducted in 2017 which stated that, comparing babies born of VVD and counterparts (VBD group) were more likely to have fetal distress and about fivefold as likely to suffer from birth asphyxia [21]. After coming head entrapment, cord accidents, and lack of experienced health workers in breech deliveries could be the main reasons that predisposed breech fetuses to an increased risk of lower Apgar in the first and fifth minutes.

This study wasn't without limitations. Being retrospective in nature, difficulty in determining timing of stillbirth, identification of fetal status during admission, and documentation of only gross congenital anomalies might influence the results of this study. The sample size was not calculated as new updated data recording system was introduced in 2019. But this option provided us a chance to involve more participants in the study rather than the sample size without much incurring us extra cost.

## Conclusion

Vaginal breech delivery was associated with low first minute Apgar. Address from others zobas, grandmultiparity, low birth weight, and multiple fetus's increases risk of having vaginal breech delivery while primiparity and increasing birth weight were protective.

## Supporting information

**S1 Checklist. PLOS One human subjects research checklist.**
(PDF)

## Acknowledgments

Authors acknowledge the Orotta Hospital Maternity staff for their cooperation in the data collection process

## Author contributions

**Conceptualization:** Okbu Frezgi, Hagos Teckle, Andebrhan Tewelde.

**Formal analysis:** Okbu Frezgi, Berhe Tesfai, Abraham Amanuel, Khalid Hussein, Hagos Teckle, Hailemichael Gebremariam, Andebrhan Tewelde.

**Methodology:** Okbu Frezgi, Berhe Tesfai, Abraham Amanuel, Khalid Hussein, Hailemichael Gebremariam, Andebrhan Tewelde.

**Visualization:** Okbu Frezgi.

**Writing – original draft:** Okbu Frezgi.

**Writing – review & editing:** Okbu Frezgi, Berhe Tesfai, Abraham Amanuel, Khalid Hussein, Hagos Teckle, Hailemichael Gebremariam, Andebrhan Tewelde.

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
