## [Decision Letter · Decision Letter 0]

18 Mar 2025

Dear Dr. Frezgi,

**There are some major methodological flaws identified by one of the reviewers that needs to be addressed before we can consider this manuscript for publication. Please address the concerns on the way the data was analysed. **

Please submit your revised manuscript by May 02 2025 11:59PM. If you will need more time than this to complete your revisions, please reply to this message or contact the journal office at plosone@plos.org . A rebuttal letter that responds to each point raised by the academic editor and reviewer(s). You should upload this letter as a separate file labeled 'Response to Reviewers'.A marked-up copy of your manuscript that highlights changes made to the original version. You should upload this as a separate file labeled 'Revised Manuscript with Track Changes'.An unmarked version of your revised paper without tracked changes. You should upload this as a separate file labeled 'Manuscript'.

We look forward to receiving your revised manuscript.

Kind regards,

Mergan Naidoo, PhD

Academic Editor

PLOS ONE

**Journal Requirements:**

1. When submitting your revision, we need you to address these additional requirements. Please ensure that your manuscript meets PLOS ONE's style requirements, including those for file naming. The PLOS ONE style templates can be found at https://journals.plos.org/plosone/s/file?id=wjVg/PLOSOne_formatting_sample_main_body.pdf and https://journals.plos.org/plosone/s/file?id=ba62/PLOSOne_formatting_sample_title_authors_affiliations.pdf 2. When completing the data availability statement of the submission form, you indicated that you will make your data available on acceptance. We strongly recommend all authors decide on a data sharing plan before acceptance, as the process can be lengthy and hold up publication timelines. Please note that, though access restrictions are acceptable now, your entire data will need to be made freely accessible if your manuscript is accepted for publication. This policy applies to all data except where public deposition would breach compliance with the protocol approved by your research ethics board. If you are unable to adhere to our open data policy, please kindly revise your statement to explain your reasoning and we will seek the editor's input on an exemption. Please be assured that, once you have provided your new statement, the assessment of your exemption will not hold up the peer review process. 3. Please amend either the abstract on the online submission form (via Edit Submission) or the abstract in the manuscript so that they are identical.

Reviewers' comments:

Reviewer's Responses to Questions

**Comments to the Author**

1. Is the manuscript technically sound, and do the data support the conclusions?

Reviewer #1: No

Reviewer #2: Yes

2. Has the statistical analysis been performed appropriately and rigorously?

Reviewer #1: No

Reviewer #2: Yes

3. Have the authors made all data underlying the findings in their manuscript fully available?

Reviewer #1: Yes

Reviewer #2: Yes

4. Is the manuscript presented in an intelligible fashion and written in standard English?

Reviewer #1: No

Reviewer #2: Yes

**Reviewer #1:**  Thank you for the opportunity to review the submission, and for conducting this important piece of research.

I think this article needs lots of work prior to meeting PLoS publication criteria.

Scientific method

- Research question is not clearly answered

- A couple of results mentioned in Discussion [200-1] do not Appear in Results

English language

- Use of perinatal vs. neonatal

- uniformity of CD vs. CS

- provide definitions of abbreviations (COR, AOR)

- what is VCD [100]

- univariable vs. univariate

- provide definitions of abortion and extremely premature

- risk vs. likelihood

Statistics

- Difficult to interpret in places (especially multivariate analyses - unclear what is being isolated to highlight effect of exposure of interest)

- No sample size calculation

- Unclear why 1:2 ratio of case:control

- Median is mentioned (how is it relevant)

- Mean should be presented with SD

- it may have been better to exclude multiple pregnancies

Ethics

- I think the ethics statement in the proforma should refer to the approval by the Research Ethics committee

Obtetric issues

- Unclear if maternal morbidity referred to in [64] is immediate vs. long-term. Also unclear what the nature of maternal morbidity related to VBD is

- What is your reason for using the birth strata you used

- Can you comment on the significance of the decrease in VBD over the study period, and the potential reason?

Citations

- Many listed as invalid

Limitations

- I feel there are more than are mentioned

**Reviewer #2:**  Thank you for an interesting study.

The results need to be noted.

What I missed it the fact whether the breech deliveries ("cases") had a foetal heart on admission in labour or not. i.e. Were all fresh stillbirths. That should be confirmed and discussed at the relevant positions.

In the discussion I also missed the argument that the increased rate of stillbirth may be related to the relatively poor experience of vaginal breech deliveries of the current clinicians?

**Do you want your identity to be public for this peer review?** For information about this choice, including consent withdrawal, please see our Privacy Policy

Reviewer #1: **Yes: ** Adam Konrad Asghar

Reviewer #2: No

---

## [Author Response · Author response to Decision Letter 1]

24 Apr 2025

Thank you for your interesting comments really helped us to reshape our article.

---

## [Decision Letter · Decision Letter 1]

5 Jun 2025

Dear Dr. Frezgi,

Thank you for submitting your revised manuscript to PLOS ONE. After careful consideration, we feel that it has merit but does not fully meet PLOS ONE’s publication criteria as it currently stands. Therefore, we invite you to submit a revised version of the manuscript that addresses the points raised during the review process.

We look forward to receiving your revised manuscript.

Kind regards,

Mergan Naidoo, PhD

Academic Editor

PLOS ONE

Journal Requirements:

Reviewers' comments:

Reviewer's Responses to Questions

**Comments to the Author**

Reviewer #1: (No Response)

2. Is the manuscript technically sound, and do the data support the conclusions?

Reviewer #1: Yes

3. Has the statistical analysis been performed appropriately and rigorously?

Reviewer #1: Yes

4. Have the authors made all data underlying the findings in their manuscript fully available?

Reviewer #1: Yes

5. Is the manuscript presented in an intelligible fashion and written in standard English?

Reviewer #1: Yes

Reviewer #1: Thank you for a detailed rebuttal.

I would suggest minor revisions as follows:

1. See lines 85/86. As previously suggested, use perinatal OR neonatal. Title now says Perinatal, but neonatal is used twice in these lines.

2. Choose either multivariable or multivariate https://pmc.ncbi.nlm.nih.gov/articles/PMC3518362/

3. Suggest listing a lack of a sample size calculation as a limitation. Same with decision to use 1:2 case:control. Your argument makes logical sense, but we are not sure if it makes statistical sense.

4. Median is still mentioned (line 110)

**Do you want your identity to be public for this peer review?** For information about this choice, including consent withdrawal, please see our Privacy Policy

Reviewer #1: **Yes: ** Adam Konrad Asghar

---

## [Author Response · Author response to Decision Letter 2]

16 Jul 2025

Reviewers comments Authors response

Reviewer #1

1. See lines 85/86. As previously suggested, use perinatal OR neonatal. Title now says Perinatal, but neonatal is used twice in these lines. Thank you. Arrangements have being made.

2. Choose either multivariable or multivariate https://pmc.ncbi.nlm.nih.gov/articles/PMC3518362/

Thank you. Arrangements have being made.

3. Suggest listing a lack of a sample size calculation as a limitation. Same with decision to use 1:2 case: control. Your argument makes logical sense, but we are not sure if it makes statistical sense. Added in limitation part with explanation.

But, the choice of 1:2 ratio of cases to controls was based on the idea that it was easy to recruit cases and controls with no extra cost, and the outcome was considered more common event. For this reason we didn’t add in limitation part.

4. Median is still mentioned (line 110)

Arrangements have being made.

---

## [Editor Report · Decision Letter 2]

29 Jul 2025

Dear Dr. Frezgi,

1. See lines 85/86. As previously suggested, use perinatal OR neonatal. Title now says Perinatal, but neonatal is used twice in these lines.

2. Choose either multivariable or multivariate https://pmc.ncbi.nlm.nih.gov/articles/PMC3518362/

3. Suggest listing a lack of a sample size calculation as a limitation. Same with decision to use 1:2 case:control. Your argument makes logical sense, but we are not sure if it makes statistical sense.

We look forward to receiving your revised manuscript.

Kind regards,

Mergan Naidoo, PhD

Academic Editor

PLOS ONE
---

## [Author Response · Author response to Decision Letter 3]

26 Aug 2025

Reviewers comments Authors response

Reviewer #1

1. See lines 85/86. As previously suggested, use perinatal OR neonatal. Title now says Perinatal, but neonatal is used twice in these lines. Thank you. Change have being made.

2. Choose either multivariable or multivariate https://pmc.ncbi.nlm.nih.gov/articles/PMC3518362/

Thank you. Arrangements have being made.

3. Suggest listing a lack of a sample size calculation as a limitation. Same with decision to use 1:2 case: control. Your argument makes logical sense, but we are not sure if it makes statistical sense. Added in limitation part with explanation.

But, the choice of 1:2 ratio of cases to controls was based on the idea that it was easy to recruit cases and controls with no extra cost, and the outcome was considered more common event. For this reason we didn’t add in limitation part.

4. Median is still mentioned (line 110)

Change have being made.

---

## [Editor Report · Decision Letter 3]

10 Sep 2025

Perinatal Outcome of Vaginal Breech Delivery in Orotta National Referral Teaching Hospital, Eritrea, 2024; a Case Control Study.

PONE-D-24-59072R3

Dear Dr. Okbu Frezgi

We’re pleased to inform you that your manuscript has been judged scientifically suitable for publication and will be formally accepted for publication once it meets all outstanding technical requirements.

Kind regards,

Mergan Naidoo, PhD

Academic Editor

PLOS ONE
---

## [Editor Report · Acceptance letter]

PONE-D-24-59072R3

PLOS ONE

Dear Dr. Frezgi,

I'm pleased to inform you that your manuscript has been deemed suitable for publication in PLOS ONE. Congratulations! Your manuscript is now being handed over to our production team.

Kind regards,

on behalf of

Professor Mergan Naidoo

Academic Editor

PLOS ONE